# Apelin Promotes Prostate Cancer Metastasis by Downregulating TIMP2 via Increases in miR-106a-5p Expression

**DOI:** 10.3390/cells11203285

**Published:** 2022-10-19

**Authors:** Tien-Huang Lin, Sunny Li-Yun Chang, Pham Minh Khanh, Nguyen Thi Nha Trang, Shan-Chi Liu, Hsiao-Chi Tsai, An-Chen Chang, Jo-Yu Lin, Po-Chun Chen, Ju-Fang Liu, Jeng-Hung Guo, Chun-Lin Liu, Hsi-Chin Wu, Chih-Hsin Tang

**Affiliations:** 1School of Post-Baccalaureate Chinese Medicine, Tzu Chi University, Hualien 970374, Taiwan; 2Department of Urology, Buddhist Tzu Chi General Hospital Taichung Branch, Taichung 427213, Taiwan; 3Graduate Institute of Biomedical Sciences, China Medical University, Taichung 403433, Taiwan; 4School of Medicine, China Medical University, Taichung 403433, Taiwan; 5Department of Medical Education and Research, China Medical University Beigang Hospital, Yunlin 651012, Taiwan; 6Translational Medicine Center, Shin Kong Wu Ho-Su Memorial Hospital, Taipei 111045, Taiwan; 7Department of Life Science, National Taiwan Normal University, Taipei 106308, Taiwan; 8School of Oral Hygiene, College of Oral Medicine, Taipei Medical University, Taipei 110301, Taiwan; 9Department of Neurosurgery, China Medical University Hospital, Taichung 404333, Taiwan; 10Department of Urology, China Medical University Hospital, Taichung 404333, Taiwan; 11Department of Urology, China Medical University Beigang Hospital, Yunlin 651012, Taiwan; 12Department of Medical Laboratory Science and Biotechnology, Asia University, Taichung 400354, Taiwan; 13Chinese Medicine Research Center, China Medical University, Taichung 404333, Taiwan

**Keywords:** prostate cancer, apelin, TIMP2, miR-106-5p, metastasis

## Abstract

Prostate cancer commonly affects the urinary tract of men and metastatic prostate cancer has a very low survival rate. Apelin belongs to the family of adipokines and is associated with cancer development and metastasis. However, the effects of apelin in prostate cancer metastasis is undetermined. Analysis of the database revealed a positive correlation between apelin level with the progression and metastasis of prostate cancer patients. Apelin treatment facilitates cell migration and invasion through inhibiting tissue inhibitor of metalloproteinase 2 (TIMP2) expression. The increasing miR-106a-5p synthesis via c-Src/PI3K/Akt signaling pathway is controlled in apelin-regulated TIMP2 production and cell motility. Importantly, apelin blockade inhibits prostate cancer metastasis in the orthotopic mouse model. Thus, apelin is a promising therapeutic target for curing metastatic prostate cancer.

## 1. Introduction

Prostate cancer commonly affects the urinary tract of men, whose susceptibility for this cancer increases with age [1]. Around 77% of newly diagnosed prostate cancer cases present with localized disease; 11% and 5% of cases present with disease that has metastasized to the regional lymph nodes and distant organs, respectively [2]. Five-year survival is around 30% in patients with metastatic prostate cancer, whereas almost 100% of patients with local or regional disease remain alive at 5 years [3]. A clear understanding of the mechanism of malignant progression and metastasis in prostate cancer would enable the development of early prevention and intervention strategies.

Tumor metastasis involves various processes, such as reorganizing and degrading the extracellular matrix (ECM) barrier to allow cancer cell growth, invasion and migration. Matrix metalloproteases (MMPs) are zinc-dependent proteases [4,5] that are key mediators of tumor motility [6], so inhibiting MMP activation is crucial for inhibiting cancer metastasis [7,8]. Tissue inhibitors of metalloproteinases (TIMPs) are endogenous proteins that regulate MMP production and thus prohibit tumor development and metastasis. TIMP1, 2 and 3 all belongs to the TIMP family [9]. TIMP1 regulates the balance of matrix remodeling [10] and plays important biological functions in cell proliferation and metastasis [10,11]. TIMP3 possesses the ability to regulate tumor growth, metastasis, angiogenesis, and other physiological progress through controlling MMPs production [12]. In addition, TIMP3 methylation has been documented in several malignancies [13]. High levels of TIMP2 synthesis predict better survival in both endometrial cancer [14] and pancreatic cancer [15], and TIMP2 reportedly regulates colorectal cancer metastasis [16]. In prostate cancer, the inhibition of TIMP2 expression enhances ECM degradation, promoting the motility and metastasis of prostate cancer cells [17]. TIMP2 is therefore an important target in the management of cancer metastasis.

Apelin belongs to the adipokine family [18]. Apelin is important for normal physiological processes such as metabolism [19], angiogenesis [20], proliferation, and motility [21]. Apelin is also closely associated with the progression of many cancers [22,23]. Overexpression of apelin enhances the motility and migration of lung cancers [24], hepatocellular cancers [25] and colon cancers [26]. Although, the effect of apelin upon prostate cancers metastasis is unknown. Our investigation found higher levels of apelin expression in human prostate cancer tissue than in normal healthy samples. Apelin also appeared to promote the migration and invasion of prostate cancer cells by inhibiting TIMP2 production. Increasing miR-106a-5p synthesis via the c-Src/PI3K/Akt pathway regulated the effects of apelin, indicating that apelin may be worth targeting in metastatic prostate cancer.

## 2. Materials and Methods

### 2.1. Materials

Antibodies against apelin, TIMP2, c-Src, p85, Akt and β-actin were purchased from GeneTex (Hsinchu, Taiwan). c-Src, p85, Akt and control siRNAs were obtained from Dharmacon (Lafayette, CO, USA). Antibodies against p-c-Src, p-p85 and p-Akt were purchased from Cell Signaling Technology (Danvers, MA, USA). PP2, Ly294002 and Akt inhibitors were bought from Sigma-Aldrich (St. Louis, MO, USA).

### 2.2. Cell Culture

All human prostate cancer cell lines were obtained from the American Type Culture Collection (Manassas, VA, USA). Cells were cultured in RPMI-1640 medium (Gibco BRL, Rockville, MD, USA) with 10% FBS and placed in a humid sterile incubator at 37 °C and 5% CO_2_.

### 2.3. Real-Time Quantitative PCR Analysis of mRNA and miRNA

Total RNA was isolated from prostate cancer cells using TRIzol reagent (MDBio; Taipei, Taiwan). RNA (1 μg) was reverse-transcribed into cDNA with oligo-DT primer, according to the manufacturer’s protocol (Invitrogen; Carlsbad, CA, USA). qPCR was performed using SYBR Green with sequence-specific primers. qPCR assays were performed with StepOne Software v2.3 and StepOnePlus sequence detection system (Applied Biosystems, Waltham, MA, USA) [27,28].

### 2.4. Western Blot

Prostate cancer cells were applied with RIPA buffer. Isolated proteins were applied to SDS-PAGE and transferred to PVDF membranes (Merck; Darmstadt, Germany) [29,30]. The membranes were blocked with 5% non-fat milk, applied with primary antibodies, then washed and applied with secondary antibodies. The membranes were visualized by the ImageQuant™ LAS 4000 biomolecular imager (GE Healthcare, Chicago, IL, USA) [31,32,33].

### 2.5. Study Datasets

Levels of apelin, TIMP2 and miRNA in healthy and tumor tissue samples collected from The Cancer Genome Atlas (TCGA) were analyzed by the Gene Expression Profiling Interactive Analysis (GEPIA), and the Oncomine and UALCAN websites [34].

### 2.6. Transwell Assay

Cell migratory and invasive activity was evaluated through Transwell inserts in 24-well dishes (Costar, New York, NY, USA) mentioned in our previous research [35]. Migratory and invasive cells were imaged under ×200 magnification using an Eclipse Ti2 microscope (Nikon, Tokyo, Japan).

### 2.7. Transient Transfection and Luciferase Assays

PC3 and DU145 cells were transfected with Src, p85, and Akt siRNAs, apelin shRNA, or pCDNA3.1-APLN and pCDNA3.1-TIMP2 plasmids. The mixed DNA and Lipofectamine 2000 were transfected to the cells for 24 h.

Luciferase activity was assayed using the method described in our previous publications [36]. Briefly, after 24 h transfection, the lysis buffer (Promega, Madison, WI, USA) was added, and 100 μL was added to each well. Cell lysate were collected and an equal volume of luciferase substrate was added into cell lysates, relative luciferase activity was determined by Dual-Luciferase^®^ Reporter Assay System (Promega, Madison, WI, USA).

### 2.8. Metastatic Prostate Cancer Model

All procedures in the animal studies were approved by the Institutional Animal Care and Use Committee, and performed according to the Guidelines of Animal Experimentation issued by China Medical University. Six-week old male nude mice, bought from BioLASCO Taiwan Co., Ltd. (Taipei, Taiwan), were injected using a 22-gauge needle in the anterior prostate with 5 × 10^5^ PC3 or PC3/sh-APLN cells suspended in 50 μL Matrigel. Tumor development and distal metastasis was monitored using the IVIS Spectrum scanner (Xenogen, Tucson, AZ, USA). Mice were humanely sacrificed after 7 weeks and distant organs (liver and lungs, as well as leg bone), were harvested. The percentages of lung metastases (No. of positive signal lungs/No. of total lungs) and other metastases were conducted using the IVIS Imaging System. Hematoxylin and eosin (H&E) staining was used to investigate distal metastasis in bone, liver and lung specimens.

### 2.9. Immunohistochemistry (IHC)

Human and mouse tissues were stained with anti-apelin or TIMP2 antibody and the procedure as according our previous works [37,38]. The sum of the intensity and percentage scores were analyzed [34].

### 2.10. Statistics

All values are given as the mean ± standard deviation (S.D.). The Student’s *t*-test assessed the differences between the groups. A *p* value of <0.05 was considered to be statistically significant.

## 3. Results

### 3.1. Apelin Is Highly Expressed in Patients with Progressing Prostate Cancer or Metastatic Disease

Apelin promotes cancer progression and metastasis [22,23]. Our analysis of apelin mRNA expressions in samples from the TCGA database revealed higher apelin expression in prostate cancer samples than in normal healthy tissue (Figure 1A) and apelin levels were positively correlated with T and N tumor classification, as well as Gleason grade in the prostate cancer samples (Figure 1B). Consistently, analyses of the Oncomine database and GEPIA tool, as well as IHC data from the tissue array, confirmed significantly upregulated apelin expression in prostate cancer samples (Figure 1C–E) and higher levels of apelin were negatively associated with survival (Figure 1F). Interestingly, apelin expression was higher in metastatic prostate cancer (androgen receptor [AR] positive) compared to nonmetastatic disease (Figure 1G). Our results indicate that high levels of apelin are positively related with increasing tumor stage, poor survival and metastasis in prostate cancer.

### 3.2. Apelin Facilitates Prostate Cancer Cell Motility by Reducing TIMP2 Expression

To examine the effects of apelin in prostate cancer cell motility, the PC3 and DU145 cell lines were stimulated with apelin. The results showed that apelin concentration dependently enhances cell migration and invasion (Figure 2A,B) but did not influence viability of PC3 and DU145 cells (Appendix A). A previous report has indicated higher migratory ability of PC3 cells compared to DU145 and LNCaP cells [39]. Our data indicated higher levels of apelin expression in PC3 cells than in DU145 and LNCaP cells (Figure 2C), implying that apelin is associated with migratory capacity in prostate cancer lines. Overexpression in LNCaP cells (lowest expressed apelin) or knockdown of apelin in PC3 cells (highest expressed apelin) promoted and reduced prostate cancer cell motility, respectively (Figure 2D–I), suggesting that apelin facilitates prostate cancer migration and invasion.

TIMPs inhibit MMPs synthesis and suppress tumor metastasis in different cancer types [40]. Stimulation of PC3 cells with apelin significantly inhibited TIMP2 synthesis by a greater extent than TIMP1 and TIMP3 synthesis (Figure 3A). Apelin also concentration-dependently decreased mRNA and protein synthesis of TIMP2 (Figure 3B,C). Furthermore, knockdown apelin increases the TIMP2 expression (Appendix A). Overexpression of TIMP2 antagonized apelin-induced promotion of cell motility (Figure 3D–F). The analysis of TCGA and GEPIA records as well as IHC data from the tissue array also revealed lower levels of TIMP2 in prostate cancer tissue than in normal healthy specimens (Figure 3G–I). Thus, apelin facilitates prostate cancer disease by inhibiting TIMP2 expression.

### 3.3. Apelin Suppresses TIMP2 Synthesis and Promotes Prostate Cancer Cell Motility by Increasing miR-106a-5p Expression

miRNAs are crucial modulators in the regulation of cancer progression and metastasis [41]. When we combined the miRwalk (http://mirwalk.umm.uni-heidelberg.de/interactions/, accessed on 8 August 2022) and ONCO.IO (http://onco.io/node.php?nid=88300, accessed on 8 August 2022) databases to predict which miRNAs could bind most effectively with TIMP2, 3 miRNAS were identified (miR-20a, miR-93a, and miR-106a-5p) (Figure 4A). In response to apelin treatment, miR-106a-5p synthesis was markedly enhanced compared with either miR-20a or miR-93 (Figure 4B). In addition, incubation of apelin with PC3 or DU145 cells increased miR-106a-5p production in a concentration-dependent manner (Figure 4C). Knockdown apelin decreases the miR-106a-5p expression (Appendix A). Cells were applied with the miR-106a-5p inhibitor reversed apelin-induced regulation of TIMP2 expression and prostate cancer cell motility (Figure 4D–F). To further examine whether miR-106a-5p specially binds to TIMP2, we established luciferase reporter vectors containing the wild-type or mutant 3′-UTRs of TIMP2 mRNA (Figure 4G). Apelin inhibited wild-type but not mutant TIMP2 3′-UTR luciferase activity (Figure 4H). The level of miR-106a-5p was higher in prostate cancer patients than in normal individuals (Figure 4I). These results show that apelin promotes prostate cancer cell motility by inhibiting TIMP2 expression via increasing miR-106a-5p synthesis.

### 3.4. Apelin Inhibits TIMP2 Expression and Subsequently increases Prostate Cancer Cell Motility by Promoting miR-106a-5p Synthesis via the c-Src/PI3K/Akt Signaling

c-Src is involved in several key signaling pathways and plays an important role in tumorigenesis and metastasis [42]. Treatment of prostate cancer cells with the c-Src inhibitor (PP2) or siRNA antagonized apelin-regulated effects on TIMP2 expression, cell migration and invasion (Figure 5A–F). Apelin stimulation enhanced c-Src phosphorylation (Figure 5G). The c-Src inhibitor and siRNA both antagonized apelin-induced increases in miR-106a-5p synthesis (Figure 5H,I). c-Src activation of PI3K/Akt pathway is crucial for cancer metastasis [43]. Treating prostate cancer cells with PI3K and Akt inhibitors or transfecting them with PI3K and Akt siRNAs reversed apelin-mediated effects upon TIMP2 synthesis, cell motility and miR-106a-5p synthesis (Figure 6A–H). Apelin also promoted the phosphorylation of PI3K and Akt, while the c-Src and PI3K inhibitors diminished apelin-induced promotion of PI3K and Akt phosphorylation (Figure 6I–K). Knockdown apelin decreases the c-Src, PI3K and Akt phosphorylation (Appendix A). Thus, apelin induces TIMP2-dependent prostate cancer motility by increasing miR-106a-5p expression via the c-Src/PI3K/Akt pathway.

### 3.5. Apelin Blockade Inhibits Prostate Cancer Metastasis in the Orthotopic Model

Next, we addressed whether apelin blockade helps to prevent prostate cancer metastasis in tumor xenograft mouse models. First, we implanted PC-3 cells stably expressing pLenti CMV V5-Luc into nude mouse anterior prostates, then sacrificed the mice 7 weeks later (Figure 7A). Apelin blockade significantly suppressed tumor growth, according to IVIS imaging and manual tumor weight measurements (Figure 7A–C). Importantly, apelin blockade also prevented distant metastases in the lung, liver and bone (Figure 7D,E). IHC staining demonstrated that apelin blockade significantly suppressed levels of apelin expression and increased levels of TIMP2 (Figure 7E). These results support the targeting of apelin for reducing the development of metastatic prostate cancer. Figure 8 represents the overall study results.

## 4. Discussion

It is well established that advanced prostate cancer is related with aggressive growth and a high metastatic likelihood [44]. The dysfunction of apelin, a cytokine secreted from adipose tissue, plays a role in tumor metastasis by enhancing cell proliferation, migration and cell survival [45]. This study reports high levels of apelin expression in human metastatic prostate cancer samples. Upregulation of apelin expression predicts a poor prognosis with lower survival in patients with prostate cancer. Apelin facilitates TIMP2-dependent migration and invasion of prostate cancer cells. Elevating miR-106a-5p expression via the c-Src/PI3K/Akt signaling cascades was mediated in apelin-induced promotion of prostate cancer motility. Importantly, apelin blockade inhibited prostate cancer metastasis in the orthotopic mouse model. The evidence indicates that apelin is a promising therapeutic object in metastatic prostate cancer.

Secretion of androgen by testicular Leydig cells plays a pivotal role in prostate epithelial cell growth and differentiation [46], as well as prostate cancer development and metastasis [47]. The requirement for androgen in prostate cancer progression led to the facilitation of androgen deprivation therapy [48]. In this study, apelin overexpression clearly enhanced the motility of androgen-dependent prostate cancer cells (the LNCaP cell line). Exogenous apelin significantly enhanced the migration and invasion of two androgen-independent cell lines (PC3 and DU145), while shRNA knockdown of apelin autocrine inhibited c-Src/PI3K/Akt signaling pathway (Appendix A) and motility of PC3 and DU145 cells. Epithelial-mesenchymal transition (EMT) involves changes in epithelial cells to mesenchymal cells [49] and associated with tumor cell invasion and cancer metastasis [50,51]. However, our results (Appendix A) show that apelin did not significantly change the EMT markers expression. These results indicate that apelin facilitates the migration and invasion but not EMT function in both androgen-dependent and -independent prostate cancer cells.

TIMPs are well recognized for their inhibition of MMPs, which promote tumor metastasis [16]. In this study, stimulation of prostate cancer cells with apelin abolished TIMP2 expression by a greater extent than TIMP1 or TIMP3, suggesting that TIMP2 is more important than other TIMPs in apelin-mediated motility of prostate cancer cells. TIMP2 regulates metastasis in several cancers, including cervical [52], lung [53] and ovarian cancer [54]. Our study analyses indicate that compared with normal controls, prostate cancer patients have lower levels of TIMP2 expression. We observed that apelin promoted prostate cancer cell migration and invasion that was antagonized when the cells were transfected with TIMP2 overexpression. These results suggest that apelin facilitates TIMP2-dependent prostate cancer cell motility.

The APJ receptor is a major receptor of apelin. The apelin/APJ system is a critical regulator of various physiological functions, such as glycometabolism, liver disease and macrophage activation [55,56,57]. Here we also found that the APJ expression is consistence with aplein levels, which is highest expressed in PC3 cells and lowest expressed in LNCaP cells (Appendix A), associating with apelin facilitates the migration and invasion ability. c-Src pathway is critical for controlling different cellular functions [58], such as the facilitation of cell invasion and metastasis [59,60]. In current study found that apelin increases c-Src phosphorylation, while the c-Src inhibitor or siRNA reversed apelin-regulated TIMP2 synthesis and the motility of prostate cancer cells. PI3K/Akt signaling is crucial downstream of c-Src for regulating the migratory ability of human chondrosarcoma cells during metastasis [61]. Our data showed that the PI3K and Akt inhibitor or siRNA antagonized apelin-mediated TIMP2-dependent prostate cancer motility. Our results also reveal that apelin enhances PI3K and Akt phosphorylation, which was antagonized by the c-Src inhibitor, suggesting that c-Src-dependent PI3K/Akt activation mediates apelin-controlled TIMP2 synthesis and the metastasis of human prostate cancer cells.

The distinguishing feature of miRNAs is their ability to target several genes and affect multiple biological functions, including cancer [62,63]. Numerous reports indicated that pharmacotherapy capable of modulating miRNA expression would antagonize cancer cell migration and thus be an novel therapeutic approach for tumor metastasis [64,65]. In current study, our investigation of miRNA software databases predicted that miR-106a-5p interrupts with TIMP2 transcription. This was confirmed by the results revealing higher miR-106a-5p level in prostate cancer tissue than in normal control tissue. In addition, apelin stimulation increases miR-106a-5p expression, while treatment of prostate cancer cells with the miR-106a-5p inhibitor reversed apelin-induced inhibition of TIMP2 synthesis and promotion of cell migration and invasion. c-Src, PI3K and Akt inhibitors, as well as their respective siRNAs, inhibited apelin-facilitated promotion of miR-106a-5p synthesis, suggesting that apelin inhibits TIMP2 synthesis and facilitates prostate cancer migration by increasing miR-106a-5p production through the c-Src/PI3K/Akt pathway. Our in vivo studies also showed that knockdown apelin inhibited miR-106a-5p expression (Appendix A) and the nodular metastasis in the lives and bones of nude mice versus negative control groups, which is consistence with our in vitro experiments (Appendix A) and previous report [66].

## 5. Conclusions

Apelin suppresses TIMP2 production and subsequently facilitates the metastatic potential of human prostate cancer cells by increasing miR-106a-5p synthesis via the c-Src, PI3K and Akt signaling cascades. We now have a better understanding about how apelin-mediated miRNA and TIMP2 synthesis contributes to prostate cancer cell motility, which may help investigators design more effective treatment for metastatic disease.

## Figures and Tables

**Figure 1 cells-11-03285-f001:**
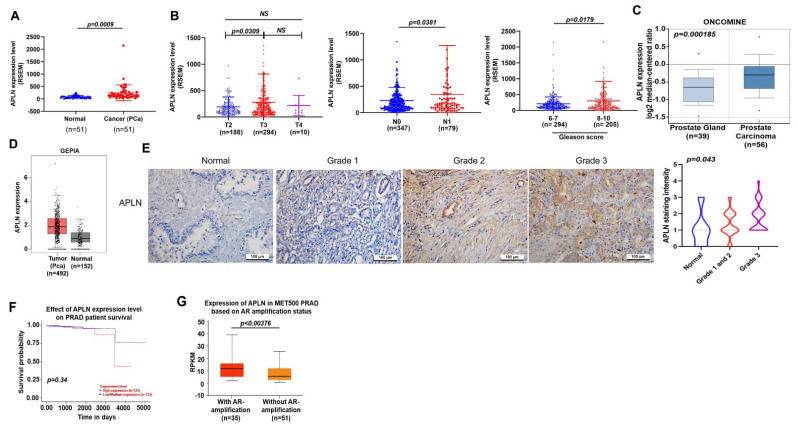
Apelin levels are associated with clinicopathological characteristics in prostate cancer metastasis. (**A**,**B**) Apelin mRNA levels in prostate cancer tissues were examined by the TCGA for tumor volume, lymph node metastasis and Gleason scores. Dots of bar graph means the number of patient. (**C**) Apelin levels in the prostate gland and cancerous prostate tissue from the Oncomine database. (**D**) Apelin expression in prostate cancer tissue samples and normal healthy tissue from the GEPIA database. (**E**) Representative images of IHC staining for apelin in tissue samples from healthy individuals and prostate cancer patients. (**F**) Increasingly lower survival time was associated with increasing levels of apelin expression. (**G**) Metastatic prostate cancer exhibited higher levels of apelin expression compared to primary prostate cancer (exposed by AR amplification). Scale bar for all images is 100 μm. NS, no significant difference.

**Figure 2 cells-11-03285-f002:**
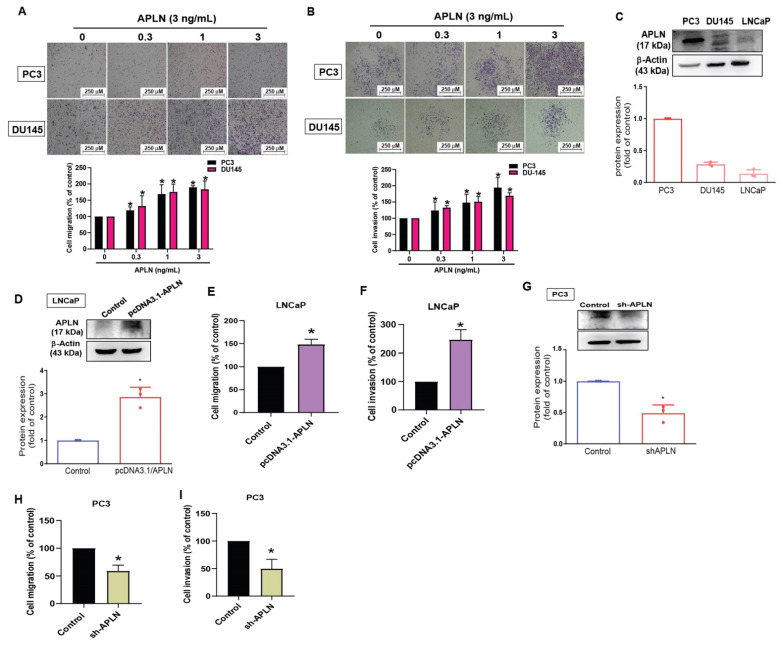
Apelin facilitates the migration and invasion of prostate cancer cells. (**A**,**B**) Prostate cancer cells were treated with apelin for 24 h, then migration (*n* = 4) and invasion (*n* = 4) was performed by the Transwell assay. (**C**) Apelin levels in indicated cells were investigated by Western blot (*n* = 3). (**D**–**F**) LNCaP cells were transfected with apelin cDNA, then cell migration (*n* = 4), invasion (*n* = 4) and levels of apelin synthesis (*n* = 3) were performed. (**G**–**I**) PC3 cells were transfected with apelin shRNA, then cell migration (*n* = 4), invasion (*n* = 4) and apelin expression (*n* = 3) were performed. * *p* < 0.05 compared with the control group. Scale bar for all images is 250 μm. Dots of bar graph means the repeat number.

**Figure 3 cells-11-03285-f003:**
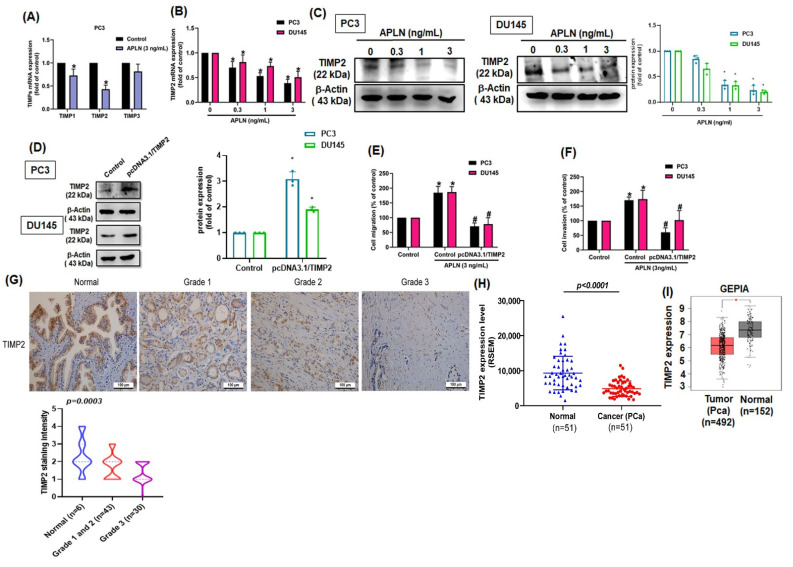
Apelin promotes prostate cancer motility by inhibiting TIMP2 expression. (**A**) PC3 cells were treated with apelin, then TIMP expression was examined by qPCR (*n* = 4). (**B**,**C**) Cells were treated with apelin for 24 h, then TIMP2 expression was examined by qPCR (*n* = 4) and Western blot (*n* = 3). (**D**–**F**) Cells were transfected with TIMP2 cDNA followed by stimulation with apelin, then cell migration (*n* = 4), invasion (*n* = 4) and levels of TIMP2 expression (*n* = 3) were examined. (**G**) Representative images of IHC staining for TIMP2 in tissue samples from healthy individuals and prostate cancer patients. (**H**,**I**) Tissue samples from the TCGA and GEPIA databases were measured for levels of TIMP2 in normal and prostate cancer tissue samples. Dots of bar graph means the number of patient. * *p* < 0.05 compared with the control group; # *p* < 0.05 compared with the apelin-treated group. Scale bar for all images is 100 μm.

**Figure 4 cells-11-03285-f004:**
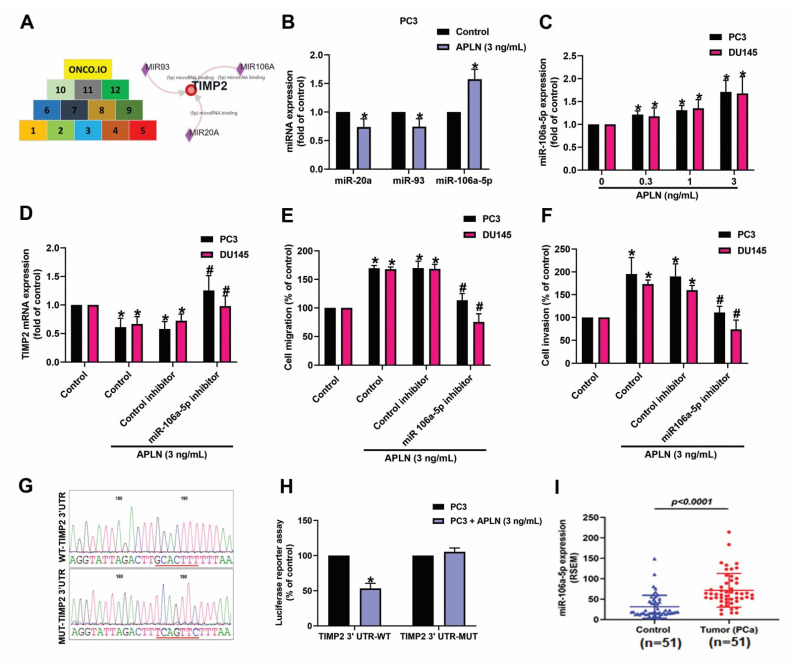
Apelin decreases TIMP2 expression and promotes prostate cancer cell motility by increasing miR-106a-5p expression. (**A**) MiRNA interference with TIMP2 transcription was predicted by combining 12 datasets from both miRwalk and ONCO.IO. (**B**,**C**) Cells were treated with apelin, then miRNA expression was examined by qPCR (*n* = 4). (**D**–**F**) Cells were applied with the miR-106a-5p inhibitor followed by stimulation with apelin, then cell migration (*n* = 4), invasion (*n* = 4) and levels of TIMP2 expression (*n* = 3) were performed. (**G**) Schematic 3′-UTR representation of the TIMP2 sequence. Red line indicated the point mutation site of miR-105a binding to TIMP2 sequence. (**H**) PC3 cells were applied with indicated plasmid, then treated with apelin and luciferase activity (*n* = 3) was measured. (**I**) Tissue samples from the TCGA database were measured for levels of miR-106a-5p expression in normal and prostate cancer tissues. Dots of bar graph means the number of patient. * *p* < 0.05 compared with the control group; # *p* < 0.05 compared with the apelin-treated group.

**Figure 5 cells-11-03285-f005:**
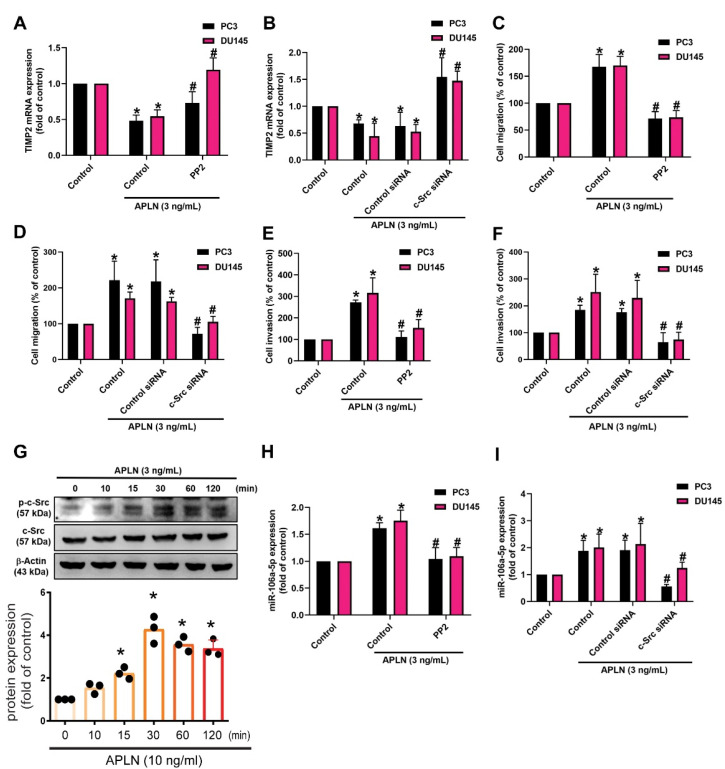
The c-Src pathway is mediated in apelin-induced migration and invasion of prostate cancer cells. (**A**–**F**,**H**,**I**) Cells were applied with the c-Src inhibitor (PP2; 1 μM) or siRNA, then incubated with apelin, before examining cell migration (*n* = 4) and invasion (*n* = 4), TIMP-2 and miR-106a-5p expression (*n* = 4). (**G**) PC3 cells were treated with apelin, then c-Src phosphorylation was investigated by Western blot (up-panel) and the quantify data were provide in low panel (*n* = 3). * *p* < 0.05 compared with the control group; # *p* < 0.05 compared with the apelin-treated group. Dots of bar graph means the repeat number of Western blot assay.

**Figure 6 cells-11-03285-f006:**
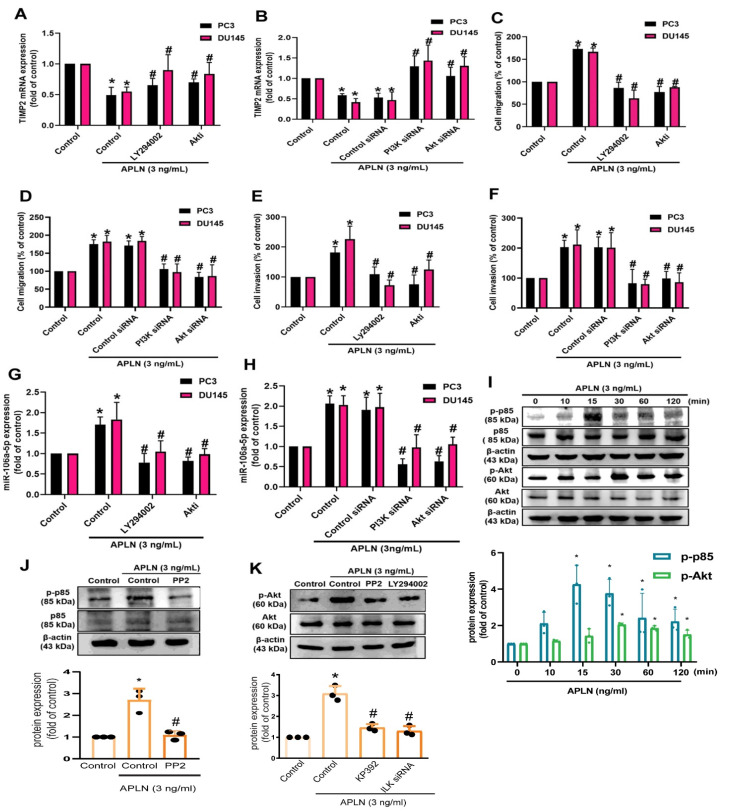
The PI3K/Akt pathway is involved in apelin-induced motility of prostate cancer cells. (**A**–**H**) Cells were applied with inhibitors of PI3K (LY294002; 2 μM) and Akt (2 μM), or transfected with PI3K and Akt siRNAs, then applied with apelin, before examining cell migration (*n* = 4) and invasion (*n* = 4), TIMP2 and miR-106a-5p expression (*n* = 4). (**I**–**K**) PC3 cells were treated with apelin or pretreated with c-Src and PI3K inhibitors then applied with apelin for 30 min, before examining PI3K and Akt phosphorylation by Western blot (*n* = 3). * *p* < 0.05 compared with the control group; # *p* < 0.05 compared with the apelin-treated group. Dots of bar graph means the repeat number of Western blot assay.

**Figure 7 cells-11-03285-f007:**
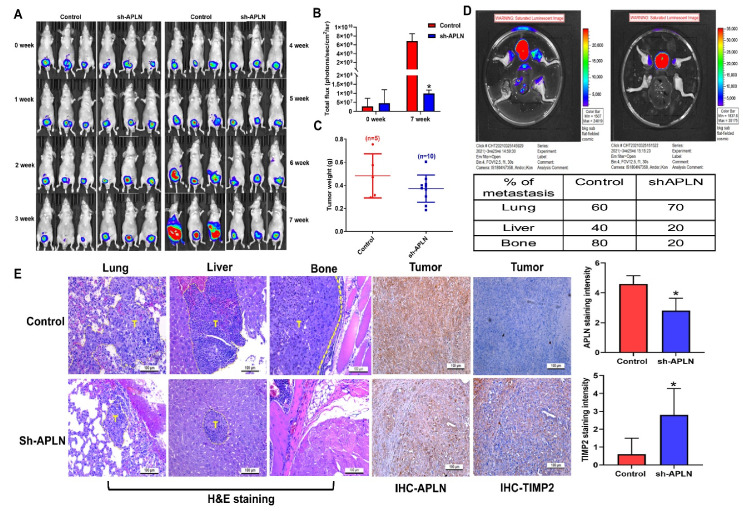
Apelin blockade inhibits prostate cancer metastasis in vivo. (**A**–**E**) PC3-Luc (*n* = 6) or PC3/sh-APLN-Luc cells (*n* = 6) were injected into the ventral prostates of SCID mice. Tumor growth, tumor metastasis to lungs, livers and legs were investigated by IVIS System. (**E**) Tissue specimens were stained with apelin or TIMP2 antibody and applied to IHC analysis. * *p* < 0.05 compared with the control group; Scale bar for all images is 100 μm.

**Figure 8 cells-11-03285-f008:**
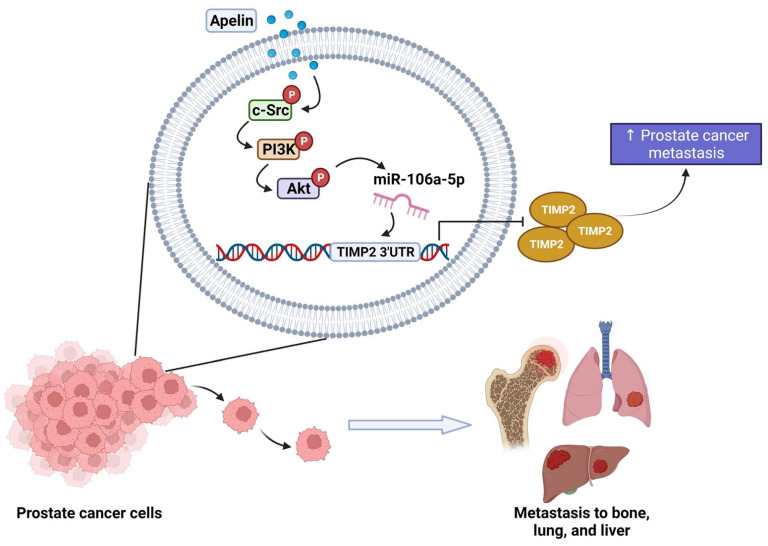
Schematic diagram summarizes the mechanisms by which apelin facilitates metastasis in human prostate cancer cells. Apelin suppresses TIMP2 production and subsequently facilitates the metastatic potential of human prostate cancer cells by increasing miR-106a-5p production via the c-Src, PI3K and Akt signaling pathways.

## Data Availability

The data generated and analyzed will be made from the corresponding author on reasonable request.

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
