# Peer review of "Apelin Promotes Prostate Cancer Metastasis by Downregulating TIMP2 via Increases in miR-106a-5p Expression"

_cells, 2022, doi:10.3390/cells11203285_

Round 1

Reviewer 1 Report

In this manuscript, the authors studied whether and how apelin would promote prostate cancer development and metastasis. By comparing the prostate cancer tissues with the normal healthy ones, the authors found significant difference in the expression level of apelin and when exposed to apelin, the migration and invasion of prostate cancer lines were dramatically promoted via the inhibition of TIMP2 synthesis. Further analyses of the public dataset and in vitro mechanistic studies identified miR-106pa-5P as the mediator to regulate the TIMP2 expression via the c-Src/PI3K/Akt signaling pathways. The authors then verified their theory in a xenograft mouse model by inhibiting the prostate cancer metastasis through apelin blockade. Overall, this is a well-designed study that revealed a novel role of apelin in prostate cancer. However, with some important experimental details and critical analyses missing, it's difficult for this reviewer to decide the validity of some of the results and conclusions before the following issues can be addressed:   

The authors should add some background information on the TIMP family in, especially those that are studied in the manuscript, like TIMP1 and 3. 

What are the sample sizes for Figure 1 (A), Figure 3 (H), Figure 4(I) and Figure 7?

Statistical analysis is missing from Figure 1(D)

The IHC images in Figure 1(E) and 3(G) are too blur to read, the authors should provide pictures with better resolution. Besides, how many tissue samples are used in scoring the staining intensity?

Details about the Luciferase reporter assay are missing from the method part

what are the cell viability of PC3 and DU145 under different concentrations of apelin? What are the reasons these two cell lines were selected? And why western blot analyses were only done in the PC3 line?

The number of repeats for every experiment should be mentioned in the method and and figure legends.

Every western blot image should be accompanied by a quantification and statistical analysis

Why was the apelin overexpression done in the LNCap cell line in figure 1(D) but not in DU145, considering it's the major research target in the rest of the manuscript.

The basal expression of TIMP2 in PC3 seems to be significantly different in Figure 3(C) and (D)

The western blot image of TIMP2 overexpression in DU145 is missing from Figure 3 (D)

How long were the samples treated in Figure 6(J) and (K)? In Figure 6(J), it doesn't look like the expression of p-p85 was stimulated by APLN in the control group compared to the non-treated control group. And also, in Figure 6(K), as the inhibitor of c-Src, pp2 should decrease p-Akt through inhibiting the phosphorylation of c-src, however, the expression of p-Akt in the APLN/PP2 treated group dose not look less than that of the APLN control group.

What are the numbers in Figure 7(D)? How are they calculated?

In the animal study, what are the status of miR-160a-5p and c-Src/PIK3/Akt signaling pathway in the control vs. the shAPLN group? Are their expression in consistent with the proposed mechanism in the schematic diagram in Figure 8? If not, what could be the reasons?

Author Response

Reviewer 1:

Q1: The authors should add some background information on the TIMP family in, especially those that are studied in the manuscript, like TIMP1 and 3.

A: More information has accordingly been provided, as follows: “TIMP1, 2 and 3 all belongs to the TIMP family [9]. TIMP1 regulates the balance of matrix remodeling [10] and plays important biological functions in cell proliferation and metastasis [10, 11]. TIMP3 possesses the ability to regulate tumor growth, metastasis, angiogenesis, and other physiological progress through controlling MMPs production [12]. In addition, TIMP3 methylation has been documented in several malignancies [13].” (Lines 54-58).

Q2: What are the sample sizes for Figure 1 (A), Figure 3 (H), Figure 4(I) and Figure 7?

A: The sizes all provided in Figure 1 (A), Figure 3 (H), Figure 4(I) and Figure 7

Q3: Statistical analysis is missing from Figure 1(D)

A: The statistical analysis of Figure 1(D) has been provided.

Q4: The IHC images in Figure 1(E) and 3(G) are too blur to read, the authors should provide pictures with better resolution. Besides, how many tissue samples are used in scoring the staining intensity?

A: Higher-quality images are provided for Figures 1E and 3G. The tissue sample also added in the Figures 1E and 3G.

Q5: Details about the Luciferase reporter assay are missing from the method part.

A: More detailed information has been provided in the Methods “2.7. Transient transfection and luciferase assays” section, as follows:

“Luciferase activity was assayed using the method described in our previous publications [37]. Briefly, after 24 h transfection, the lysis buffer (Promega, Madison, WI) were added 100 ml to each well. Cell lysate were collected and an equal volume of luciferase substrate added into cell lysates, relative luciferase activity was determined by Dual-Luciferase® Reporter Assay System (Promega, Madison, WI, USA).” (Lines 116-120)

Q6: (i) what are the cell viability of PC3 and DU145 under different concentrations of apelin? (ii)What are the reasons these two cell lines were selected? (iii)And why western blot analyses were only done in the PC3 line?

A: (i) The cell viability of PC3 and DU145 under different concentrations didn’t any effect in MTT assay. More information has now been provided, as follows:

“The results showed that apelin concentration-dependently enhances cell migration and invasion (Fig. 2A&B) but did not influence viability of PC3 and DU145 cells (Supple-mental Fig. 1).” (Lines 171-172)

(ii) The majority of human prostate cancer cell lines, including the two "classical" cell lines PC-3 and DU-145, are reported to be androgen receptor (AR)-negative cell line. A previous report has indicated higher migratory ability of PC3 cells compared with DU145. So this two cell lines were selected for experiments.

(iii) Most of our important key data (migration, invasion, and TIMP2 expression assays) were used the two cell lines for experiments to confirm the apelin regulated c-Src/PIK3/Akt signaling pathway or promoted TIMP2 or apelin levels in these two cells. In the phosphorylation of signaling molecules, we used PC3 (the most common prostate cancer cell line) to present the activation of signaling pathway.

Q7: The number of repeats for every experiment should be mentioned in the method and and figure legends.

A: The number of repeats for every experiment has been adding in figure legends.

Q7: Every western blot image should be accompanied by a quantification and statistical analysis.

A: We have accordingly provided statistical analysis in all Western blots.

Q8: Why was the apelin overexpression done in the LNCap cell line in figure 2(D) but not in DU145, considering it's the major research target in the rest of the manuscript.

A: Thanks for reviewer attention, because the western blot in figure 2C shows that the apelin were lowest expressed in LNCap cell line. To make sure the apelin levels is associated with cell migration and invasion, the lowest expressed apelin cell line (LNCaP cells) was used to overexpression apelin, as follows:    

“Our data indicated higher levels of apelin expression in PC3 cells than in DU145 and LNCaP cells (Fig. 2C), implying that apelin is associated with migratory ability in prostate cancer cells. Overexpression in LNCaP cells (lowest expressed apelin) or knockdown of apelin in PC3 cells (highest expressed apelin) promoted and reduced prostate cancer cell motility, respectively (Fig. 2D-I), suggesting that apelin facilitates prostate cancer migration and invasion.” (Lines: 173-179)

Q8: The basal expression of TIMP2 in PC3 seems to be significantly different in Figure 3(C) and (D)

A: Thank you for your attention. More experiments have been performed and the consistence result has been provided in figure 3D

Q9: The western blot image of TIMP2 overexpression in DU145 is missing from Figure 3 (D).

A: More experiments have been performed and the western blot image of TIMP2 overexpression in DU145 has been provided in figure 3D.

Q10: (i) How long were the samples treated in Figure 6(J) and (K)? (ii) In Figure 6(J), it doesn't look like the expression of p-p85 was stimulated by Apelin in the control group compared to the non-treated control group. And also, in Figure 6(K), as the inhibitor of c-Src, pp2 should decrease p-Akt through inhibiting the phosphorylation of c-src, however, the expression of p-Akt in the Apelin /PP2 treated group dose not look less than that of the Apelin control group.

A: (i) The PC3 cells treated with apelin 30 min and then further study in Figure 6(J) and (K), as follows:    

“(I-K) PC3 cells were treated with apelin or pretreated with c-Src and PI3K inhibitors then stimulated with apelin for 30 min before examining PI3K and Akt phosphorylation by Western blot (n=3).” (Lines: 271-273)

(ii) More experiments have been performed and the western blot image of p-p85 in Figure 6J and p-Akt in Figure 6K has been provided.

Q11: What are the numbers in Figure 7(D)? How are they calculated?

A: (i) The number of Figure 7(D) has been provided.

(ii) We used IVIS system to count the positive signal tissues (Lungs, livers, and legs) / total tissues to quantify the tumor metastasis percentage, as follows:

“Mice were humanely sacrificed after 7 weeks and distant organs (liver and lungs, as well as leg bone), were harvested. The percentages of lung metastases (No. of positive signal lungs/No. of total lungs) and other metastases were conducted using the IVIS Imaging System.” (Lines 130-132)

Q12: In the animal study, what are the status of miR-160a-5p and c-Src/PIK3/Akt signaling pathway in the control vs. the shAPLN group? Are their expression in consistent with the proposed mechanism in the schematic diagram in Figure 8? If not, what could be the reasons?

A: Because the tumor size is not enough to do the western blot to check the c-Src/PIK3/Akt signaling pathway in vivo. So we just can use qPCR assay to check the miR-160a-5p levels in tumor of control and the shAPLN group. We found the miR-160a-5p levels was down-regulated in shAPLN group tumor sample (Supplemental Fig. 5). The result was confirming the schematic diagram in Figure 8, as follows:

“Our in vivo studies also showed that knockdown apelin inhibited miR-106a-5p expression (Supplemental Fig. 5) and the nodular metastasis in the lives and bones of nude mice versus negative control groups, which is consistence with our in vitro experiments (Supplemental Fig. 2B.) and previous report [67].” (Lines 365-368)

Reviewer 2 Report

Authors evaluated the role of apelin in prostate cancer metastasis and the molecular pathway altered by its effect. Using basic biological assays, they reported that apelin is upregulated in tumor tissues and exerts its actions by inhibiting TIMP2 expression via increased miR-106a-5p expression as well as altering the c-Src/PI3K/Akt pathway. Authors present some interesting findings but some points need to be addressed:

COMMENTS

1)    In evaluating the role of apelin and stimulation of cells by apelin, authors made no mention of the apelin receptor. What is the level of expression of apelin receptor in PC3, DU-145 & LNCaP cells and does that impact how these cell lines distinctly responded to migration and invasion following apelin stimulation?

2)    Rather than exogenous stimulation of prostate cancer cell lines with apelin, did authors utilize cells over- or under-expressing apelin and observe altered TIMP1, miR-106a-5p and Src-PI3K-Akt pathway via western blotting?

3)    Will it be fair to say that the observed effects of apelin on TIMP1, miR-106a-5p and Akt pathway only occurs with exogenous paracrine apelin activation of cells, rather than from autocrine release of apelin by prostate cancer cells themselves?

4)    Since manuscript focuses on metastasis, does apelin alter the expression profile of other epithelial-mesenchymal transition (EMT) markers? 

Author Response

Reviewer 2:

Q1: In evaluating the role of apelin and stimulation of cells by apelin, authors made no mention of the apelin receptor. What is the level of expression of apelin receptor in PC3, DU-145 & LNCaP cells and does that impact how these cell lines distinctly responded to migration and invasion following apelin stimulation?

A: Thank you for making this good point. More experiments have been performed and provided in Supplemental Fig. 4, as follows:    

“The APJ receptor is a major receptor of apelin. The apelin/APJ system is a critical regulator of various physiological functions, such as glycometabolism, liver disease and macrophage activation [56-58]. Here we also found that the APJ expression is consistence with apelin levels, which is highest expressed in PC3 cells and lowest expressed in LNCaP cells (Supplemental Fig. 4), associating with apelin facilitates the migration and invasion ability.” (Lines: 337-341)

Q2: Rather than exogenous stimulation of prostate cancer cell lines with apelin, did authors utilize cells over- or under-expressing apelin and observe altered TIMP1, miR-106a-5p and Src-PI3K-Akt pathway via western blotting?

A: Thank you for making this good point. More experiments have been performed and provided in Supplemental Fig. 2, as follows:

“Furthermore, knockdown apelin increases the TIMP2 expression (Supplemental Fig. 2).” (Lines 194-195).

“Knockdown apelin decreases the miR-106a-5p expression (Supplemental Fig. 2).” (Lines 221-222)

“Knockdown apelin decreases the c-Src, PI3K and Akt phosphorylation (Supplemental Fig. 2).” (Lines 255-256)

Q3: Will it be fair to say that the observed effects of apelin on TIMP1, miR-106a-5p and Akt pathway only occurs with exogenous paracrine apelin activation of cells, rather than from autocrine release of apelin by prostate cancer cells themselves?

A: In our Supplemental Fig. 2 data shows that shRNA knockdown of apelin autocrine inferenced the levels of TIMP2, miR-106a-5p, and c-Src/PI3K/Akt signaling pathway, as follows:

“Exogenous apelin significantly enhanced the migration and invasion of two androgen-independent cell lines (PC3 and DU145), while shRNA knockdown of apelin autocrine inhibited c-Src/PI3K/Akt signaling pathway (Supplemental Fig. 2) and motility of PC3 and DU145 cells.” (Lines 317-320)

Q4:  Since manuscript focuses on metastasis, does apelin alter the expression profile of other epithelial-mesenchymal transition (EMT) markers?

A: More experiments have been performed and provided in Supplemental Fig. 2, as follows:

“Epithelial-mesenchymal transition (EMT) involves changes in epithelial cells to mesen-chymal cells [50] and associated with tumor cell invasion and cancer metastasis [51, 52]. However, our results (Supplemental Fig. 3) show that apelin did not significant changed the EMT markers expression. These results indicate that apelin facilitates the migration and invasion but not EMT function in both androgen-dependent and -independent pros-tate cancer cells.” (Lines 320-325s).

Round 2

Reviewer 1 Report

The authors have done a great job addressing all concerns this reviewer raised in the review report and no further revisions in the experimental part are needed though some grammar checking is required. For example, in line 323, "significant" should be "significantly"; in line 365, "knockdown apelin" could be "knocking down apelin" or "apelin knockdown"; in supplementary figure S3, "significant effect" should be "significantly affect".